# Phytochemicals in Gastrointestinal Nematode Control: Pharmacokinetic–Pharmacodynamic Evaluation of the Ivermectin plus Carvone Combination

**DOI:** 10.3390/ani13081287

**Published:** 2023-04-09

**Authors:** María Victoria Miró, Livio Martins Costa-Júnior, Mercedes Lloberas, Patricia Cardozo, Carlos Lanusse, Guillermo Virkel, Adrián Lifschitz

**Affiliations:** 1Laboratorio de Farmacología, Centro de Investigación Veterinaria de Tandil (CIVETAN), UNCPBA-CICPBA-CONICET, Campus Universitario, Tandil 7000, Argentina; 2Laboratorio de Farmacología, Departamento de Fisiopatología, Facultad de Ciencias Veterinarias, Universidad Nacional del Centro de la Provincia de Buenos Aires (UNCPBA), Campus Universitario, Tandil 7000, Argentina; 3Laboratory of Parasite Control, Center for Biological and Health Sciences, Federal University of Maranhão, Av. dos Portugueses 1966, São Luis 65080-805, Brazil; 4Laboratorio de Parasitología, Instituto Nacional de Tecnología Agropecuaria (INTA), Estación Experimental, Balcarce 7620, Argentina

**Keywords:** carvone, ivermectin, nematode resistance, drug–drug interaction

## Abstract

**Simple Summary:**

In the last 50 years, limited modern drugs with new mechanisms of action have been incorporated; consequently, drug resistance has increased worldwide. Thus, the search for alternative pharmacological tools is a priority in ruminant production systems. Historically, natural products have played a relevant role in drug discovery. The anthelmintic activity of various phytochemicals has mainly been studied in vitro, whereas the in vivo characterization of their pharmacological properties is a challenge for upcoming years. The administration route and pharmaceutical formulation directly influence the drug concentration attained in the target parasites and, therefore, the resultant pharmacological effect. Here, we conducted an integrated pharmacological study to evaluate the combined administration of ivermectin and carvone to lambs infected with resistant nematodes. Although carvone showed a moderate in vivo anthelmintic effect and enhanced the ivermectin systemic availability after their coadministration in lambs, the concentrations attained in target tissues and in parasites were not sufficient for obtaining optimal efficacy. Innovative pharmaceutical formulations are necessary to establish phytochemicals as a useful pharmacological tool for controlling nematodes in ruminants.

**Abstract:**

A wide variety of plant-derived phytochemicals with anthelmintic effects have been described. Most of them have shown activity against parasites in vitro but have not been extensively explored in vivo. The aim of the current work was to study the pharmacokinetic–pharmacodynamic relationship of the combined administration of carvone (R-CNE) and ivermectin (IVM) to lambs. Three trials were conducted to evaluate the pharmacological interaction between R-CNE and IVM in lambs infected with resistant nematodes. Drug concentrations were measured in plasma, target tissues, and *H. contortus* by HPLC with fluorescent (IVM) and ultraviolet (R-CNE) detection. The effect of both compounds on parasites was estimated by the fecal egg count reduction. Coadministration with R-CNE significantly increased the plasma bioavailability of IVM. R-CNE showed a moderate anthelmintic effect, which was greater on the susceptible isolate of *H. contortus*. After the combination of R-CNE and IVM as an oral emulsion, both compounds were quantified in *H. contortus* recovered from infected lambs. However, R-CNE concentrations were much lower than those reported to achieve anthelmintic effects in the in vitro assays. Optimization of the pharmaceutical formulation, dose rate, and administration schedule is needed to take advantage of the intrinsic anthelmintic activity of phytochemicals.

## 1. Introduction

Livestock management and antiparasitic drugs are the main tools used for controlling gastrointestinal nematodes in different ruminant production systems. In the last 50 years, limited modern drugs with new mechanisms of action have been incorporated; consequently, drug resistance has increased worldwide [1,2]. The discovery and development of new drug is an expensive process which may take 12–15 years, costing around 1 billion USD [3]. Thus, the search for alternative pharmacological tools that allow a sustainable control of parasitic diseases is a priority. A wide variety of plant-derived phytochemicals with anthelmintic effects have been described [4,5]. Most of them have shown activity against parasites in vitro [6,7], but their pharmacological properties have not been extensively explored in vivo. 

Carvone (CNE) (5-isopropenyl-2-methyl-2-cyclohexenone) is a colorless or pale-yellow liquid with a characteristic mint odor, insoluble in water and glycerol, but miscible with ethanol and oils. In nature, CNE appears as two enantiomers: dextrorotatory or S(+)CNE and levorotatory or R(−)CNE. The European Food Safety Authority (EFSA) has included CNE in the list of flavoring substances allowed in the European Union since 2012. From chronic toxicity studies in rodents, it was established that CNE has no carcinogenic or genotoxic action [8]. It has been shown that CNE inhibits both GABA and nicotinic channels in mice [9,10]. The anthelmintic activity of CNE as a single molecule on *Haemonchus contortus* was verified in vitro [7,11]. Likewise, the in vitro anthelmintic efficacy was enhanced after the combination of CNE with other phytochemicals [11]. Recently, CNE formulated as nanoemulsions showed antiparasitic action against eggs, larvae, and adults of multidrug-resistant strains of *H. contortus* [12]. 

In vivo, a long-term treatment of CNE plus anethol mixed with food significantly reduced the nematode eggs in feces of lambs [13]. Furthermore, the efficacy against gastrointestinal nematodes of lambs was increased after the coadministration of CNE with abamectin compared to the administration of the macrocyclic lactone alone [14]. In this context, the aim of the current work was to study the pharmacokinetic–pharmacodynamic (PK/PD) relationship of the combined administration of R-CNE and ivermectin (IVM) to lambs.

## 2. Materials and Methods

### 2.1. Trial 1: Pharmacokinetic and Efficacy of the Coadministration of IVM and R-CNE via Different Routes

In this experiment, the coadministration of the commercial formulation of IVM via the subcutaneous route and R-CNE as an oral emulsion was evaluated in lambs. Trial 1 involved 35 Corriedale and Texel crossbred lambs (mean body weight 31 kg) naturally infected with resistant gastrointestinal nematodes. The trial was conducted in a sheep experimental unit (Estación Experimental INTA, Balcarce, Argentina) where a parasite control program based on the intensive use of antiparasitic drugs has been implemented for many years, leading to anthelmintic resistance to MLs and benzimidazoles. Animals were selected on the basis of worm egg per gram counts (EPG), using the modified McMaster technique with a sensitivity of 10 EPG [15]. Briefly, individual fecal samples were weighed (3 g) and mixed thoroughly with 57 mL of saturated salt solution (400 g of NaCl/L). Then, 2 mL of the mix was transferred with a pipette to the McMaster slide, and the number of eggs was counted under a microscope. The mean EPG count in the experimental animals was 5346 ± 6386. Animals were placed in a paddock and fed hay ad libitum, together with commercial concentrate feed. All animals had free access to water. Lambs were assigned to three experimental groups ranked by EPG count. Group A received IVM (IVOMEC^®^, Boehringer Ingelheim, Ingelheim am Rhein, Germany) (single dose of 0.2 mg/kg via subcutaneous route) (*n* = 9). Group B received R-CNE (Euma, Argentina) (100 mg/kg, three oral doses administered every 24 h) (*n* = 9). Lambs of Group C were treated with IVM (single dose of IVOMEC^®^ at 0.2 mg/kg via subcutaneous route) coadministered with R-CNE (100 mg/kg, three oral doses administered every 24 h) (*n* = 9). The first dose of R-CNE was administered 10 min before the single IVM administration; then, it was repeated every 24 h until the dosing schedule was completed. Group D included untreated lambs (*n* = 8). R-CNE was formulated as an emulsion composed of Tween-80/chitosan 1% (1:6) at a final concentration of 22% of R-CNE. Briefly, the R-CNE emulsion was prepared using the high-energy ultrasonic method. R-CNE was mixed under stirring with Tween-80 to form the oil phase. The oil phase was added dropwise to the chitosan solution while stirring, and the emulsion formed was then subjected to ultrasonic emulsification for 15 min. The stability of the emulsion in room temperature storage was corroborated by visual evaluation over a period of 14 days to check the presence of creaming or breaking. 

For the plasma disposition of IVM (groups A and C), jugular blood samples (2 mL) were collected into heparinized vacutainer tubes before treatment and at 3, 9, 24, 30, 48, 54, and 72 h, and 6, 9, and 15 days post treatment. Blood samples were centrifuged at 2000× *g* for 15 min; the recovered plasma was kept in labeled vials and stored at −20 °C until the analysis of IVM by HPLC. To evaluate the effect of the treatment on the fecal egg count (FEC) as an indirect measure of efficacy, individual fecal samples from all the lambs were collected on days −1 and 15 of treatment, and the FEC reduction (FECR) was calculated. The co-procultures were prepared with 10 g of feces from a pool of each experimental group obtained on days −1 and 15. The nematode genera and species were identified through the third-stage larvae recovered from the co-procultures [16].

### 2.2. Trial 2: PK/PD of R-CNE Oral Emulsion Administered to Lambs Infected with Susceptible H. contortus

Six Corriedale male lambs (average body weight 18 kg) were artificially infected with 2000 larvae of the *H. contortus* strain known as CEDIVE, susceptible to all commercially available anthelmintics [17]. Animals were housed in pens with a concrete floor. They had access to commercial concentrate feed and water ad libitum. At the beginning of the trial (30 days after experimental infection), the FEC was 13,953 ± 13,932. On day 0, lambs were treated with an oral emulsion of R-CNE (100 mg/kg, four oral doses administered every 24 h). The R-CNE emulsion composition was similar to that described for Trial 1. On days, 0, 1, 2, and 3 jugular blood samples were taken from lambs at 2, 4, 8, and 24 h post administration for measuring R-CNE levels in the bloodstream. The effect of R-CNE on *H. contortus* was indirectly estimated by collecting fecal samples on days −1, 7, and 14 of administration to evaluate the FECR. As an initial study, the current work measured the PK/PD relationship of R-CNE by assessing its impact on parasites through a reduction in FEC and relating this effect to the concentrations found in plasma.

### 2.3. Trial 3: Target Tissue Distribution and Efficacy of R-CNE/IVM Oral Emulsion against Resistant H. contortus in Lambs

Trial 3 involved 15 Corriedale male lambs (average body weight 24 kg) housed in pens which were artificially infected with 2000 larvae of a highly resistant isolate of *H. contortus* to IVM. This isolate was obtained in our lab [18] and was maintained under laboratory conditions through successive passages in parasite-free lambs. For controlling the resistant status of isolate, infected tracer lambs were periodically treated with a subcutaneous injection of IVM at 0.2 mg/kg. The FEC reduction in these tracers after the IVM treatment was 0%. At the beginning of the trial (30 days after experimental infection), the FEC was 3754 ± 2482. On day 0, lambs were treated with an oral emulsion containing R-CNE and IVM (100 mg/kg and 0.2 mg/kg respectively). The emulsion was composed of Tween-80/chitosan 1% (1:6) at a final concentration of 22% for R-CNE and 0.11% for IVM. At the beginning of the process, R-CNE and IVM (same commercial formulations used in Trial 1) were mixed with Tween-80 to prepare the oil phase. Then, the R-CNE + IVM emulsion was prepared as described in Trial 1. The lambs received the oral emulsion only containing R-CNE on days 1, 2, and 3. At 30 h (6 h after the administration of the second dose of R-CNE) and at 78 h (6 h after the fourth dose of R-CNE), three animals were sacrificed to evaluate the distribution of R-CNE and IVM to target tissues and *H. contortus.* Lambs were knocked down with a captive bolt (Jarvis Power Actuated Stunner, Jarvis Products Corporation, Middletown, CT, USA) and immediately exsanguinated in agreement with institutionally accepted animal welfare guidelines. Samples of blood, abomasum, small intestine (cranial jejunum), and ileum contents, as well as mucosal tissue, liver, lung, and bile, were taken following the procedures described in [19]. From the abomasum of each animal, the total mass of *H. contortus* was recovered to measure the drug concentration in the parasite. The minimum amount required to measure drug concentration in the parasites was 50 mg. The effect of the combined treatment on *H contortus* was determined on the remaining lambs by collecting fecal samples on days −1, 7, and 14 of administration to evaluate the FECR. 

### 2.4. Chromatographic Analysis

IVM was extracted from plasma, tissues, and *H. contortus* following the technique described by [19,20,21]. Briefly, 0.25 mL aliquots of plasma, 0.25 g of tissues, and 50 mg of *H. contortus* were combined with 10 ng of the internal standard compound (moxidectin) and then mixed with 1 mL of acetonitrile. The solvent–sample mixture was centrifuged at 2000× *g* for 15 min, and the supernatant was manually transferred into a tube and concentrated to dryness under a stream of nitrogen. The derivatization of IVM was performed following the technique described by [22]. IVM concentrations in plasma, tissues, and *H. contortus* were determined by HPLC using a Shimadzu 10 A HPLC system with fluorescence detection reading at 365 nm (excitation) and 475 nm (emission wavelength). Calibration curves were prepared, and correlation coefficients (r) and coefficient of variations (CV) were calculated. The linear regression lines for IVM showed correlation coefficients ≥ 0.99. The mean recoveries from plasma, tissues, and *H. contortus* were 84%, 96%, and 88%, respectively. The precision of the analytical method showed a CV between 6.63% and 10.8%. The limit of quantification was established at 0.2 ng/mL or ng/g. The detection of R-CNE in plasma and tissues was performed following an adaptation of the methodology described by [23]. Aliquots of plasma, tissues, and *H. contortus* (0.25 mL, 0.25 g, and 50 mg, respectively) were mixed with 0.5 mL of cold acetonitrile and shaken for 15 min. The solvent–sample mixture was centrifuged at 4000× *g* at 4 °C for 10 min, and the supernatant was manually transferred into a tube. In the case of tissue and nematode samples, 0.5 mL of HPLC water was added. An aliquot of 50 µL was injected into the HPLC. R-CNE concentrations were determined by HPLC using a Shimadzu 10 A HPLC system with UV detection reading at 254 nm. A C18 reversed-phase column (Kromasil, Eka Chemicals, Bohus, Sweden, 5 μm, 4.6 × 250 mm) was used for separation. The mobile phase was water/acetonitrile (40/60) pumped at a flow of 1.3 mL/min. The mean recoveries from plasma, tissues, and *H. contortus* were 92%, 87% and 89%, respectively. The precision of the analytical method showed a CV between 7.5% and 14.6%. The limit of quantification was established at 0.5 µg/mL or µg/g.

### 2.5. Data Analysis

Data are expressed as the mean ± standard deviation (SD). The plasma concentration versus time curves obtained after treatment of each animal were fitted with the PK Solutions 2.0 software (summit Research Services, Ashland, OH, USA) software. Pharmacokinetic parameters were determined using a noncompartmental model method [24]. The evaluation of the FECR in the presence of a control group (Trial 1) was calculated according to the following formula [25]:(1)$$FECR\;\left(\%\right)=100\times \left(1-\frac{T}{C}\right),$$
where T is the arithmetic mean of EPG counts in the treated group at 15 days post treatment, and C is the arithmetic mean of EPG counts in the untreated control group at 15 days post treatment. In trials in which there was no untreated control group (Trials 2 and 3), the effect of treatment was determined according to the following formula [26]:(2)$$FECR\;\left(\%\right)=100\times \left(1-\frac{T2}{T1}\right),$$
where T1 is the arithmetic mean of EPG counts on day −1, and T2 is the EPG count on days 7 and 14 post treatment. The 95% confidence intervals were calculated following Coles et al. (1992). Statistical analysis was performed using the Prism 8.0 software (GraphPad Software, San Diego, CA, USA). Depending on the experiment, statistical comparisons of drug concentrations, pharmacokinetic parameters, and EPG counts were performed using Student’s *t*-test, Mann-Whitney test, ANOVA, or Kruskal–Wallis test. The correction for multiple comparisons was performed with Bonferroni or Dunn’s test. Differences were considered statistically significant at *p*-values below 0.05. 

## 3. Results

### 3.1. Pharmacokinetics and Efficacy of the Coadministration of IVM and R-CNE via Different Routes (Trial 1)

IVM was detected in groups A (IVM) and C (IVM and R-CNE) up to 15 days post treatment. Figure 1 shows the IVM plasma concentration profiles after its administration alone or coadministration with R-CNE. Pharmacokinetic parameters are summarized in Table 1. 

It was observed that coadministration of R-CNE caused a significant increase in the IVM peak plasma concentration (Cmax), being 1.94 times higher compared to that measured after IVM alone. In turn, IVM plasma concentrations in the coadministration group were higher during the R-CNE administration period. This is reflected in the bioavailability calculated as the partial AUC (AUC_0–2.25d_), being 1.5 times higher in the IVM/R-CNE group than in the lambs which only received IVM. The EPG counts for all experimental groups on day 15 post treatment, along with the FECR and upper and lower confidence limits (95%), are shown in Figure 2. A similar FECR was observed after IVM and IVM + R-CNE administration (84%), whereas, after administration of R-CNE alone, the FECR was 46.6%. The main parasitic genera involved before treatments were *Ostertagia* spp. (54%), *Trichostrongylus* spp. (39%), and *Cooperia* spp. (7%). The proportion of genera did not change significantly after treatments.

### 3.2. PK/PD of R-CNE Oral Emulsion Administered to Lambs Infected with Susceptible H. contortus (Trial 2)

R-CNE was detected in plasma between 2 and 8 h after the first dose, between 2 and 4 h after the second and third doses, and only at 2 h post treatment after the fourth dose (Figure 3). Mean R-CNE Cmax was 5.28 ± 3.21 µg/mL. A rapid absorption R-CNE was observed, with the Tmax achieved at 2 h post treatment. There was also a rapid elimination of this compound with a mean elimination half-life of 2.33 h. A trend of detecting lower concentrations with successive doses was observed. Mean R-CNE concentrations at 2 h after the first dose were significantly higher (*p* < 0.05) compared to those observed after the fourth dose (5.83 vs. 2.21 µg/mL). The effect of R-CNE oral emulsion treatment on the sensible isolate to *H. contortus* showed a significant reduction in the EPG counts at day 7 and 14 post-treatment. The FECR was above 80% but showed a high variability according to the confidence intervals (Table 2). Although PK/PD parameters were not calculated, the rapid decline in R-CNE concentrations in plasma may account for the absence of a consistent effect against the existing parasites, possibly due to insufficient drug exposure.

### 3.3. Target Tissue Distribution and Efficacy of R-CNE/IVM Oral Emulsion against Resistant H. contortus in Lambs (Trial 3)

In this trial, the experimental animals were treated on day 0 with a fixed combination of R-CNE and IVM administered as an oral emulsion Then, the lambs were treated with three successive doses every 24 h with the oral emulsion of R-CNE alone. The R-CNE and R-CNE + IVM emulsions were stable throughout the period of 14 days. The concentrations of IVM and R-CNE were quantified in the different tissues and gastrointestinal contents at 30 h and 78 h post administration. In the adults of *H. contortus*, the quantification of both compounds was performed only in the samples obtained at 30 h after administration, since, at 78 h, the mass of nematodes recovered was not sufficient to perform the measurements (<50 mg). IVM concentrations in the target sites were significantly higher than those found in the plasma at both sampling times. Whereas, in the gastrointestinal contents, IVM concentrations were 3.2–12.5 times higher than in plasma, in the gastrointestinal mucosa, this ratio was in the range of 4.8–10. IVM reached high concentrations within the adults of *H. contortus*. The average concentration of IVM in *H. contortus* measured at 30 h post treatment was 87.0 ng/g. These concentrations reflected the concentrations obtained mainly in the abomasum (mean concentration of 108 ng/g). IVM concentrations in the intestinal segments (jejunum and ileum) were significantly higher than those observed in the abomasum, ranging from 290 to 413 ng/g at 30 h post treatment. The highest concentrations of IVM were measured in bile (811 ng/mL), reflecting its main route of elimination. There was also a high distribution in the lungs (mean concentration 97.9 ng/g). IVM concentrations in plasma and target tissues at 78 h post treatment were significantly lower compared to those at 30 h. Figure 4 summarizes the IVM concentrations measured in target tissues and in *H. contortus*.

R-CNE was quantified in the plasma, target tissues, and *H. contortus*. The concentrations in tissues were higher at 78 h post treatment (6 h after the fourth dose) compared to 30 h (6 h after the second dose), which may reflect an accumulation process. Concentrations measured in plasma were in the range of 0.5–0.88 µg/mL. Mean R-CNE concentrations in *H. contortus* adults at 30 h post treatment were 4.56 µg/g, correlating with the concentrations obtained in the abomasum. The highest concentrations were found in the abomasum and cranial jejunum (between 4 and 12 µg/g) compared to the caudal segments of the intestine (ileum), where concentrations ranged between 0.5 and 4.5 µg/g. While R-CNE concentrations were not detected in the lung tissue, the quantified concentrations were between 0.7 and 2 µg/g or µg/mL in the liver and bile. Figure 5 summarizes the R-CNE concentrations in the different target tissues and in *H. contortus*. The combined treatment of R-CNE + IVM significantly reduced the FEC on days 7 and 14 post treatment. The FECR on different days with the respective confidence intervals is shown in Table 3.

## 4. Discussion

The current study focused on describing the PK/PD interactions occurring after the combined use of a synthetic anthelmintic (IVM) and a phytochemical product (R-CNE) in nematode-infected lambs. In Trial 1, IVM was administered as a traditional formulation via the subcutaneous route, and R-CNE was administered as an oral emulsion based on Tween-80/chitosan in order to obtain more sustained concentrations. The presence of R-CNE significantly increased the bioavailability of IVM in lambs, as evidenced by the 1.94-fold increase in the Cmax (Figure 1, Table 1). The significant increment in the partial AUC_0–2.25_ reflected that enhanced IVM availability occurred during the period when R-CNE was present in the bloodstream. The enhancement of IVM’s systemic availability after its coadministration with R-CNE was greater than that recently obtained after the coadministration of abamectin with R-CNE as a pure solution [14]. The presence of R-CNE during the first 3 days could have induced a drug–drug interaction by interfering with the biliary or intestinal excretion of IVM mediated by ABC transporters, as shown after the coadministration of IVM with ketoconazole or loperamide in sheep and cattle [27,28,29]. However, this positive in vivo pharmacokinetic interaction was not reflected in the effect on nematodes. The FECR values were similar after the treatment with IVM alone or combined with R-CNE (Figure 2). The activity of IVM was recently increased in vitro in the presence of the phytochemical limonene, restoring the efficacy against resistant *H. contortus* [30]. In this trial, natural infection of lambs did not include *H. contortus* in the parasite population. The main genus which showed resistance to IVM was *Cooperia*. The administration of R-CNE alone (three doses) reduced the FEC by 46%, mainly impacting genera found in the abomasum such as *Teladorsagia* spp. according to the larval culture results. 

The evaluation of the PK/PD relationship for R-CNE was conducted in Trial 2, in lambs that were experimentally infected with IVM-susceptible *H. contortus*. Higher concentrations of R-CNE in plasma were observed in lambs after its administration as an emulsion compared to the pure solution [14], reflecting the influence of the pharmaceutical formulation. The mean Cmax was 5.28 µg/mL, and R-CNE was detected between 2 and 8 h after the first dose. Interestingly, after the second and third doses, R-CNE was detected up to 4 h post administration, whereas, after the fourth dose, it was only detected at 2 h post treatment (Figure 3). The repeated administration of R-CNE may produce an induction process, affecting its own metabolism. This induction process has been described for repeated administrations of other monoterpenes such as thymol and carvacrol [31]. The administration of this therapeutic dose schedule of R-CNE determined an FECR between 84% and 87% (Table 2). The FECR value was higher than observed in Trial 1 (46%), where the predominant genera were *Teladorsagia, Trichostrongylus*, and *Cooperia.*

In Trial 3, the combination of IVM and R-CNE was administered orally through the same emulsion as the fixed combination. After the first dose with the combination, lambs were treated with the emulsion containing only R-CNE for the next 3 days. The tissue distribution study was conducted to gain a better understanding of the uptake of both compounds at target sites. The distribution and uptake of IVM tissues of parasite location and target parasites, such as *H. contortus*, have been extensively studied [19,21,32]. In all analyzed tissues, IVM concentrations were higher at 30 h post treatment than at 78 h (Figure 4). As previously reported [20,31], IVM concentrations in the tissues of parasite location were significantly higher than those in plasma. The IVM drug level in plasma and tissues was consistent with previous reports, indicating that the administration of IVM as an oral emulsion combined with R-CNE does not affect its plasma disposition kinetics. For instance, in previous studies, the mean concentrations of IVM in adult *H. contortus* after oral administration of a commercial formulation were 102 ng/g at 24 h [32] and 74 ng/g at 72 h post administration [21]. In the current trial, the mean concentration of IVM in *H. contortus* after its oral administration as an emulsion was 87 ng/g at 30 h post treatment. The high concentrations of IVM measured in the liver, lung, and bile reflect the wide tissue distribution and the primary elimination pathway of IVM in ruminants [19]. 

There are no previous reports regarding R-CNE quantification in target tissues or nematodes, emphasizing the relevance and originality of the information obtained in this study. R-CNE was measured in all tissues of interest (except lung) and in *H. contortus* (Figure 5). In tissues, the concentrations were higher at 78 h post treatment (6 h after the fourth dose) compared to 30 h (6 h after the second dose), reflecting an accumulation process. The highest concentrations were found in the abomasum and cranial jejunum (between 3.2 and 8.81 µg/g at 30 and 78 h post treatment, respectively), compared to caudal segments of the small intestine, where the concentrations were in the range of 1.5 to 3.46 µg/g at 30 and 78 h post treatment, respectively. The higher R-CNE concentrations observed in the cranial segments of gastrointestinal tract may have accounted for the efficacy observed for parasites located in the abomasum compared to intestinal nematodes. The measured concentrations in plasma were within the range of 0.5–0.88 µg/mL, which is consistent with the results observed in Trial 2, where the mean concentration was 0.69 µg/mL at 8 h post administration. The R-CNE concentrations in *H. contortus* were 4.56 µg/g, reflecting the concentrations found in the environment where the parasite is typically located (i.e., abomasum content).

The effectiveness of this treatment was evaluated in lambs infected with a highly IVM-resistant strain, resulting in an FECR of 35–46%. Although R-CNE showed an in vivo anthelmintic effect, it is evident that the resistant *H. contortus* may require greater exposure to optimize its efficacy. The underlying causes that explain the differences in efficacy of R-CNE on susceptible and resistant *H. contortus* observed in the current study should be thoroughly investigated. Changes in the structure of glutamate-gated chloride channels were proposed as one of the resistance mechanisms of nematodes to IVM [33,34]. It was demonstrated that R-CNE is a noncompetitive inhibitor of GABA-gated chloride channels [9], with which IVM also interacts. Thus, the alterations in chloride channels in the *H. contortus* highly resistant to IVM may be influencing the action of R-CNE. Another relevant factor to understand the PK/PD relationship of R-CNE is related to the concentrations attained in vivo in the target tissues. These concentrations were much lower than those reported to achieve anthelmintic effects in the in vitro assays. *H. contortus* egg hatch and larval motility tests showed that concentrations of R-CNE exceeding 370 μg/mL are needed to achieve over 90% efficacy [11], several times higher than the levels measured in target tissues and *H. contortus* in the current work. An in vitro evaluation of the combination of R-CNE with anethole demonstrated synergistic effects on *H. contortus* egg hatching [11]. However, when this combination was administered for 45 days in vivo, it caused a significant FECR (around 50%) [13], similar to that observed in our study. Recently, he in vitro activity of an R-CNE nanoemulsion was evaluated on a multidrug-resistant strain of *H. contortus* [12]. Significant effects were observed after R-CNE nanoemulsion incubation, but the concentrations necessary for affecting eggs, larval development, and adult motility were 125 µg/mL, 1000 µg/mL, and 400 µg/mL, respectively. Although in vitro methodologies do not usually reflect in vivo drug exposure, these findings suggest that the levels required to achieve high in vivo efficacy would be much higher than those measured in the current work. Our study had some limitations such as an evaluation of the effects of R-CNE and its combination with IVM as a function of the FECR and not directly on the adult nematodes. This may have confused the interpretation of the obtained results; however, for ethical reasons, it was not possible to sacrifice many animals. Therefore, future research is needed to evaluate if R-CNE may be a narrow-spectrum anthelmintic compound (e.g., targeting nematodes in the abomasum), as well as if the sustained administration of R-CNE alone or combined with synthetic drugs induces changes in the drug metabolism and excretion process. The calculation of PK/PD parameters using data from in vivo studies will be relevant for the evaluation of the usefulness of this pharmacological tool.

## 5. Conclusions

R-CNE showed an in vivo anthelmintic effect. The R-CNE concentration profiles achieved in tissues and in target parasites were measured for the first time, emphasizing the relevance and originality of the information described here. Interestingly, R-CNE enhanced the systemic availability of IVM after their coadministration in lambs, resulting in a favorable PK interaction. Considering that the R-CNE concentration profiles attained in target nematodes were not sufficient for obtaining optimal clinical efficacy, further work is required to achieve increased nematode exposure. Optimization of the pharmaceutical formulation, dose rate, and administration schedule is needed to take advantage of its intrinsic anthelmintic activity.

## Figures and Tables

**Figure 1 animals-13-01287-f001:**
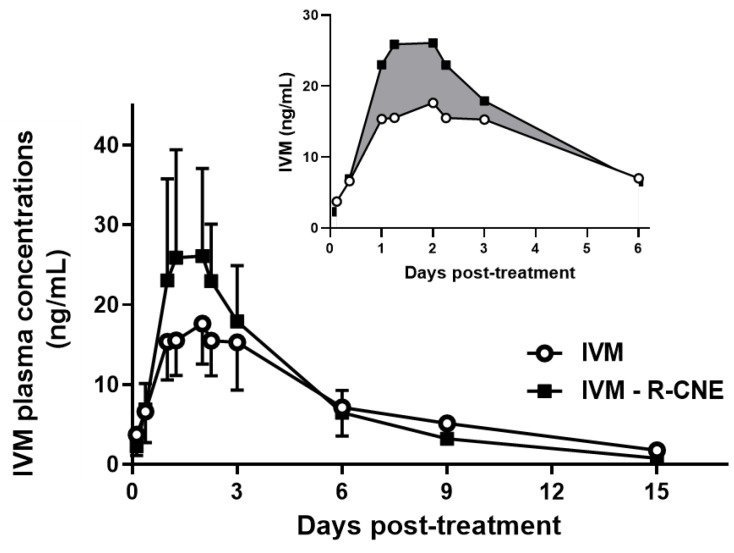
Mean plasma ivermectin (IVM) concentrations (±SD) obtained after subcutaneous administration either alone or in combination with carvone (R-CNE) (three oral doses of 100 mg/kg every 24 h) to lambs (*n* = 7 per group).

**Figure 2 animals-13-01287-f002:**
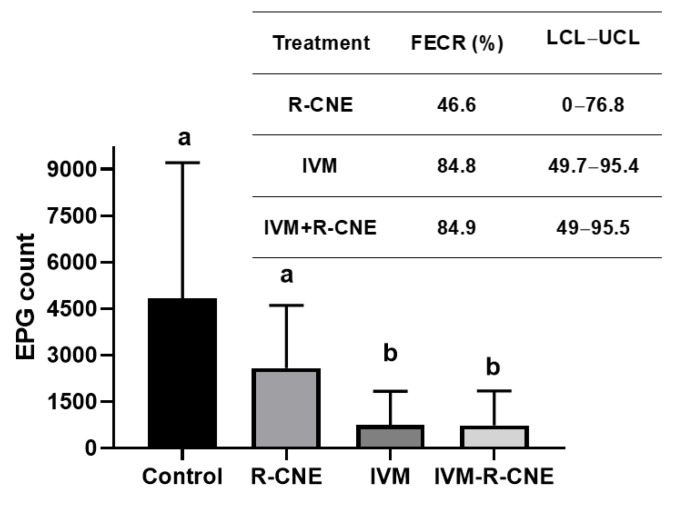
Mean (±SD) egg counts per gram of feces (EPG) obtained at 15 days for untreated control lambs (*n* = 8) and those treated subcutaneously with ivermectin (IVM) (0.2 mg/kg, *n* = 9) either alone or in combination with carvone (R-CNE) (three oral doses of 100 mg/kg every 24 h, *n* = 9). The inset shows the fecal egg count reduction (FECR) with the upper (UCL) and lower (LCL) confidence limits in lambs naturally infected with IVM-resistant nematodes. Different letters denote statistically different values compared to control group at *p* ≤ 0.05.

**Figure 3 animals-13-01287-f003:**
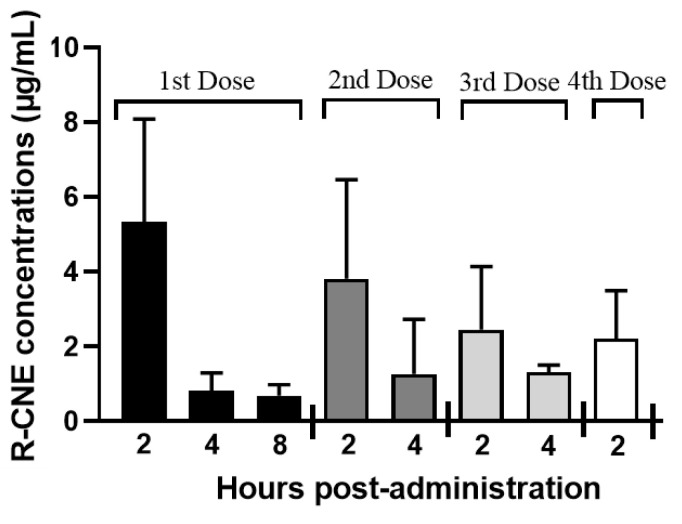
Mean (±SD) plasma carvone (R-CNE) concentrations obtained after oral administration as an emulsion (four doses of 100 mg/kg every 24 h) to lambs (*n* = 6) artificially infected with susceptible *H. contortus*. R-CNE concentrations 2 h post treatment with the first dose were significantly higher compared to those observed at 2 h post treatment with the fourth dose (*p* < 0.05).

**Figure 4 animals-13-01287-f004:**
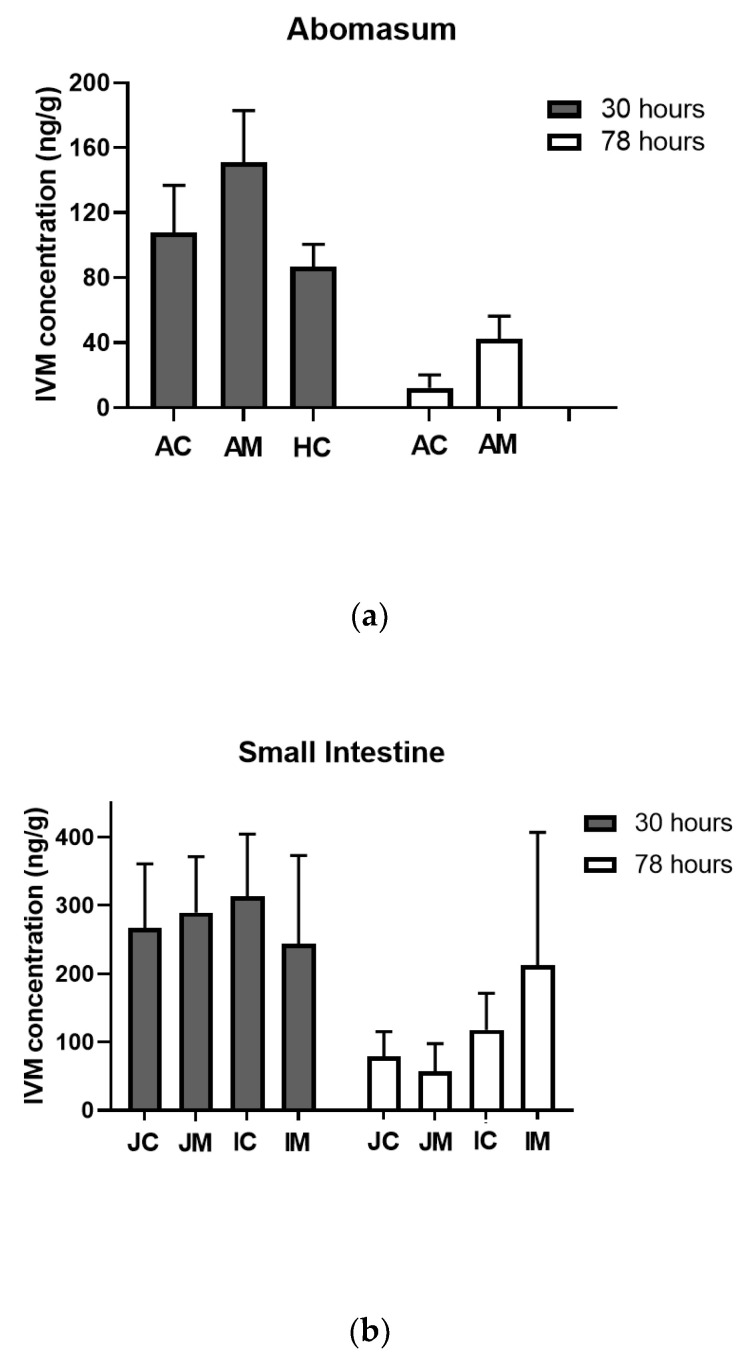
Mean (±SD) ivermectin (IVM) concentrations obtained in plasma, target tissues, and *H. contortus* after its oral administration as an emulsion (one dose, 0.2 mg/kg) combined with R-CNE (four doses every 24 h of 100 mg/kg) to lambs (*n* = 6) artificially infected with resistant *H. contortus*. (**a**) AC: abomasum content, AM: abomasum mucosa, HC: *H. contortus*; (**b**) JC: jejunum content, JM: jejunum mucosa, IC: ileum content, IM: ileum mucosa; (**c**) PL: plasma, BL: bile, LV: liver, LN: lung. IVM concentrations measured at 30 h were significantly higher compared to those measured at 78 h post treatment for AC, AM, JC, JM, IC, PL, BL, LV, and LN (*p* ≤ 0.05).

**Figure 5 animals-13-01287-f005:**
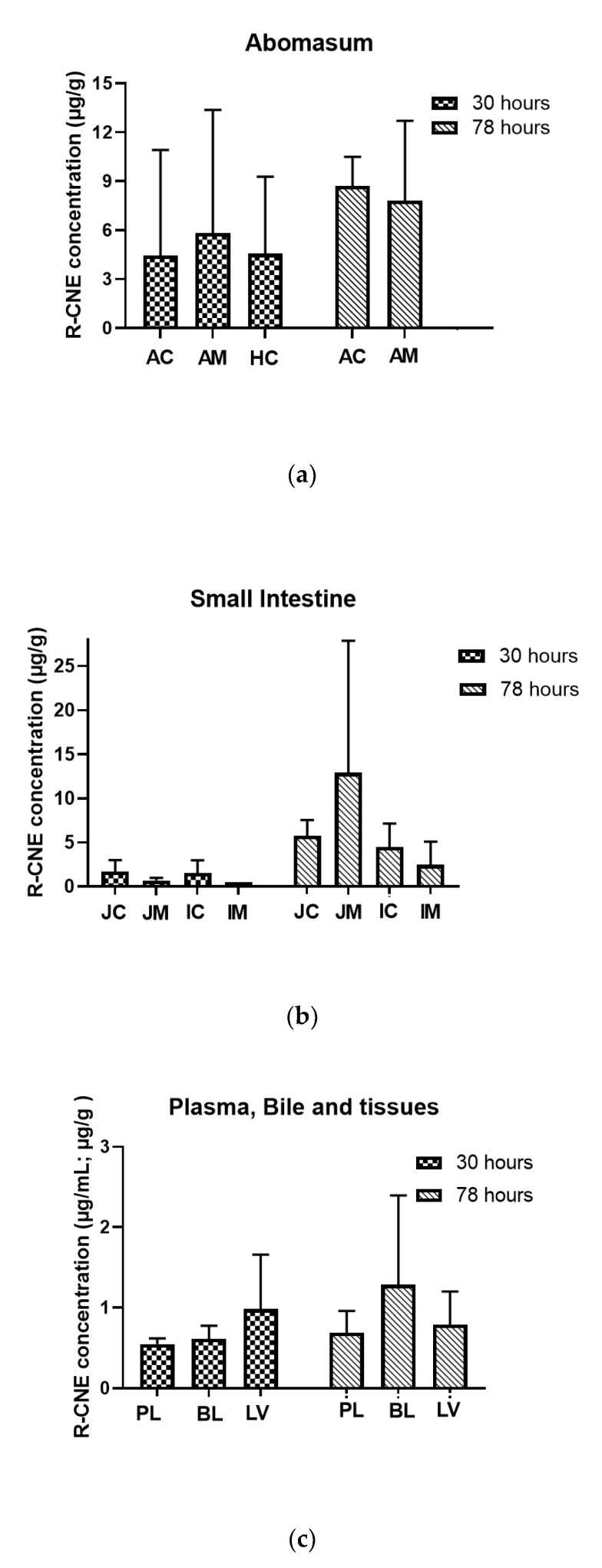
Mean (±SD) carvone (R-CNE) concentrations obtained in plasma, target tissues, and *H. contortus* after its oral administration as an emulsion (four doses every 24 h of 100 mg/kg) in a fixed combination with ivermectin (one dose, 0.2 mg/kg)) to lambs (*n* = 6) artificially infected with resistant *H. contortus*. (**a**) AC: abomasum content, AM: abomasum mucosa, HC: *H. contortus*; (**b**) JC: jejunum content, JM: jejunum mucosa, IC: ileum content, IM: ileum mucosa; (**c**) PL: plasma, BL: bile, LV: liver. R-CNE concentrations measured at 78 h were significantly higher compared to those measured at 30 h post treatment for JC, JM, IC, and IM (*p* ≤ 0.05).

**Table 1 animals-13-01287-t001:** Plasma pharmacokinetic parameters of ivermectin (IVM) in plasma (mean ± SD) obtained after subcutaneous administration (0.2 mg/kg) either alone or in combination with carvone (R-CNE) (three oral doses of 100 mg/kg every 24 h) to lambs (*n* = 7 per group).

Kinetic Parameters	IVM	IVM + R-CNE
Cmax (ng/mL)	18.4 ± 5.40 ^a^	35.7 ± 13.9 ^b^
T max (days)	2.00 ± 0.58 ^a^	1.83 ± 0.56 ^a^
AUC_0–t_ (ng d/mL)	113 ± 34.9 ^a^	121 ± 36.6 ^a^
AUC_0–2.25d_ (ng d/mL)	28.7 ± 7.93 ^a^	42.4 ± 12.3 ^b^
T ½ ab (days)	0.38 ± 0.15 ^a^	0.50 ± 0.08 ^a^
T ½ el (days)	3.29 ± 0.47 ^a^	2.52 ± 0.41 ^b^
MRT (days)	4.82 ± 1.07 ^a^	3.80 ± 0.40 ^a^

Cmax, peak plasma concentration; Tmax, time to peak plasma concentration; T ½ el, elimination half-life; AUC_0–t_, area under concentration vs. time curve from time 0 to the last concentration detected; AUC_0–2.25d_, area under concentration vs. time curve from time 0 to 2.25 days; MRT, mean residence time. Different letters between treatments for values of kinetic parameters denote a statistically significant difference at *p* ≤ 0.05.

**Table 2 animals-13-01287-t002:** Mean egg per gram (EPG) counts (±SD) and fecal egg count reduction percentage (FECR) obtained at 7 and 14 days after the oral administration of R-CNE as an emulsion (four doses of 100 mg/kg every 24 h) to lambs (*n* = 6) artificially infected with susceptible *H. contortus*.

Day	EPG	FECR (%) (LCI-UCL)
0	13,953 ± 13,932 ^a^	-
7	2170 ± 2725 ^b^	84.5 (40.5–95.9)
14	1706 ± 2249 ^b^	87.8 (51.3–96.9)

LCL: lower confidence limit, UCL: upper confidence limit. Different letters across different days denote statistically different values (*p* ≤ 0.05).

**Table 3 animals-13-01287-t003:** Mean egg per gram (EPG) counts (±SD) and fecal egg count reduction percentage (FECR) obtained at 7 and 14 days after the oral administration of an emulsion containing R-CNE + IVM (100 mg/kg and 0.2 mg/kg, respectively) on day 0 and the oral administration of R-CNE alone as an emulsion on days 1, 2, and 3 (100 mg/kg every 24 h) to lambs (*n* = 9) artificially infected with resistant *H. contortus*.

Day	EPG	FECR (%) (LCI-UCL)
0	3407 ± 2715 ^a^	-
7	1832 ± 2068 ^b^	46.2 (0–78)
14	2192 ± 2417 ^b^	35.7 (0–73.3)

LCL: lower confidence limit, UCL: upper confidence limit. Different letters among different days denote statistically different values at *p* ≤ 0.05.

## Data Availability

The data presented in this study are available on request from the corresponding author.

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
