# Peer review of "Phytochemicals in Gastrointestinal Nematode Control: Pharmacokinetic–Pharmacodynamic Evaluation of the Ivermectin plus Carvone Combination"

_animals, 2023, doi:10.3390/ani13081287_

Round 1

Reviewer 1 Report

This manuscript is an interesting well-performed study with good data presentation. 

The title of the manuscript is adequate and correctly describes the findings: “Phytochemicals on gastrointestinal nematodes control: pharmacokinetic - pharmacodynamic evaluation of the ivermectin plus carvone combination”. 

The objective of this study was to study the pharmacokinetic-pharmacodynamic (PK/PD) relationship of the combined administration of carvone (R-CNE) and ivermectin (IVM) to lambs through 3 trials:

Trial 1: Pharmacokinetic and efficacy of the co-administration of IVM and R-CNE by different routes.

Trial 2: PK/PD of R-CNE oral emulsion administered to lambs infected with susceptible H. contortus.

Trial 3: Target tissue distribution and efficacy of R-CNE-IVM oral emulsion against resistant H. contortus in lambs.

The results of the study are discussed with current references. A scientific result has been obtained and suggestions for the future are presented.

However, I have some comments and  questions:

Line 80: CNA OR CNE???

Line 81: 35 or 36 lambs were used? So far there are 28 treated animals, 9 from GA, 9 from GB, and 10 from GC. If you used 8 animals without treatment (control group), it gives a total of 36 animals, not 35. Please review

Line 110: Please, describe in more detail the methodology applied to establish the PK/PD relationship.

Line 117: Why did the animals in trial N°2 receive 4 doses of R-CNE and the animals in trial N°2 received only 3 doses?

Line 132: How was the R-CNE emulsion combined with IVM prepared for oral administration? Was the stability of this combination evaluated? Was commercial IVM used? Was an IVM formulated for oral administration to lambs used? or was the same IVM used as in trial No. 1? It is well known that bioavailability may be affected after administration by a different route. Please, clarify the procedure performed, because is not clear.

Line 228: In Figure 2, in the R-CNE legend, the hyphen between 0.7 - 6.8 is missing

Line 234: Nothing is mentioned in the results about the PK/PD efficacy predictors, and whether they were achieved. Please, add something

Figure 4 legend, line 286: replace AM with JM

Line 344: Replace CNA with CNE

Line 390: Replace "de" with "the"

Line 407: Delete one point

Line 421: Replace fist with first

Congratulations on your interesting study!

Author Response

Thank you very much for evaluating our manuscript, entitled Phytochemicals on gastrointestinal nematodes control: pharmacokinetic-pharmacodynamic evaluation of the ivermectin plus carvone combination”. We have considered all issues mentioned in the reviewers’ comments

1) Line 80: CNA OR CNE???

Thank you very much for the observation. It is R-CNE.

2) Line 81: 35 or 36 lambs were used? So far there are 28 treated animals, 9 from GA, 9 from GB, and 10 from GC. If you used 8 animals without treatment (control group), it gives a total of 36 animals, not 35. Please review

We apologize for the mistake, the total number of animals used in this experiment was 35 (three groups of 9  treated lambs each and one control group of 8 animals)

3) Line 110: Please, describe in more detail the methodology applied to establish the PK/PD relationship.

The current understanding of the anthelmintic effect of phytochemicals is primarily based on in vitro assays using conventional dose-effect analysis. However, deeper insights into the effect of these compounds can be gained by conducting in vivo studies, where the effect on parasites (PD) is correlated with the observed concentrations in plasma and target tissues (PK). As an initial stage, the current work measured the PK/PD relationship of IVM and R-CNE by assessing their impact on parasites through a reduction in egg count in feces and relating this effect to the concentrations found in plasma and parasite localization sites. Future studies should involve calculating parameters such as AUC/MIC or time above MIC, among others, to better reflect this PK/PD relationship. Advancing our understanding of the pharmacological behavior of phytochemical derivatives is needed and we believe that the current work supplies novel knowledge on this issue. We have included a comment in the manuscript.

4) Line 117: Why did the animals in trial N°2 receive 4 doses of R-CNE and the animals in trial N°2 received only 3 doses?

While administering a single dose to large animals may be ideal from a practical standpoint, the necessity to achieve higher exposure levels led us to adopt a multidose treatment approach with R-CNE in our study. Following the observation of a relatively low FECR after the three doses of R-CNE in Trial 1, we administered four doses of R-CNE to the lambs in Trials 2 and 3. Currently, we are exploring alternative dosing schedules to enhance the anthelmintic effect of phytochemicals.

5) Line 132: How was the R-CNE emulsion combined with IVM prepared for oral administration? Was the stability of this combination evaluated? Was commercial IVM used? Was an IVM formulated for oral administration to lambs used? or was the same IVM used as in trial No. 1? It is well known that bioavailability may be affected after administration by a different route. Please, clarify the procedure performed, because is not clear.

The procedure performed to prepare the emulsion was included in the manuscript. Briefly, R-CNE or R-CNE +IVM emulsions were prepared following the high-energy ultrasonic method. R-CNE or R-CNE+ IVM were mixed under stirring with Tween 80 forming the oil phase. IVM formulation used to prepare the combined emulsion was the same commercial formulation used in Trial 1. The oil phase was added dropwise to the chitosan solution while stirring and the emulsion formed was then subjected to ultrasonic emulsification for 15 min. The stability of the emulsions at room temperature was corroborated by the visual evaluation during a period of 14 days to check the presence of creaming or breaking.

6) Line 228: In Figure 2, in the R-CNE legend, the hyphen between 0.7 - 6.8 is missing

The LCL and UCL for R-CNE in Figure 2 were 0-76.8

7) Line 234: Nothing is mentioned in the results about the PK/PD efficacy predictors, and whether they were achieved. Please, add something

Although PK/PD parameters were not calculated, the rapid decline in R-CNE concentrations in plasma may account for the absence of a consistent effect against the existing parasites, possibly due to insufficient drug exposure. This comment has been included in the manuscript.

8) Figure 4 legend, line 286: replace AM with JM

It was replaced.

9) Line 344: Replace CNA with CNE

It was replaced.

10) Line 390: Replace "de" with "the"

It was replaced.

11) Line 407: Delete one point

It was deleted.

12) Line 421: Replace fist with first

It was replaced.

13) Congratulations on your interesting study!

Thank you very much for your suggestions and comments. We really appreciate them.

Reviewer 2 Report

This is an interesting paper which evaluates the effect of ivermectin plus carvone on Haemonchus contortus in lambs.

Indicate on each figure/table and in the text which trial the data come from. As presented it is confusing to the reader. Add n number of lambs used in legend for each figure/table for each trial/experiment. Indicate level of significance in figures 3,4,5 legends.

Trial 1: states 35 lambs used but if ad 8/8//9/10, comes to 36, please correct. Line 87, explain modified McMaster technique.

Table 2: explain LCI and UCS; in legend authors mention LCL and UCL.

line 274/5: are these values maximum or averages?

Figure 4: label A,B,C.

The second figure 4 (line 304) should be figure 5. In figure legend authors have LN: lung but no lung data in "C"; also label this figure A,B,C.

line 353: In Trial 3, the combination etc

Author Response

We have considered all issues mentioned in the reviewers’ comments and the whole manuscript was carefully revised and modified according to the reviewer´s suggestions. 

  • This is an interesting paper which evaluates the effect of ivermectin plus carvone on Haemonchus contortus in lambs.

We really appreciate your comments.

  • Indicate on each figure/table and in the text which trial the data come from. As presented it is confusing to the reader. Add n number of lambs used in legend for each figure/table for each trial/experiment. Indicate level of significance in figures 3,4,5 legends.

In the results section, each Trial was indicated in relationship with the presented data. The number of lambs and the significance were indicated in each Figure and Table where correspond.

  • Trial 1: states 35 lambs used but if ad 8/8//9/10, comes to 36, please correct. Line 87, explain modified McMaster technique.

We apologize for the mistake. Trial 1 involved 35 lambs (8/9/9/9)

The explanation of the modified McMaster technique was added to the text manuscript.

  • Table 2: explain LCI and UCS; in legend authors mention LCL and UCL.

We apologize for the mistake. It was corrected in Table 2 and Table 3. The correct abbreviations are LCL and UCL

  • line 274/5: are these values maximum or averages?

In the original submitted version, the value was the mean concentration of IVM in bile at 30 hours post-treatment 578 ng/mL. In R1 this value was replaced by the maximum (811 ng/mL).

  • Figure 4: label A,B,C.

The second figure 4 (line 304) should be figure 5. In figure legend authors have LN: lung but no lung data in "C"; also label this figure A,B,C.

The labels A, B, C were added to Figures 4 and 5. The mistakes observed by the Reviewer in the legends of Figure 5 were corrected. Thank you very much for correcting our mistakes

  • line 353: In Trial 3, the combination etc

 It was changed.

Reviewer 3 Report

This study deals with a topic of interest. It discusses the intrinsic anthelmintic activity of carvone and demonstrates that its coadministration significantly increases the plasma bioavailability of IVM. Moreover, it exhibits carvone concentration profiles achieved in tissues and in target parasites for the fist time. The manuscript is well written and i would like to suggest only some minor corrections:

-in the abstract define the HPLC by which drug concentrations were measured in plasma, target tissues and in H. contortus

-spp is spp. and should not be in italics

-line 80 – delete full stop

-line 390 – correct

- line 407 – delete full stop

Finally, please go through the manuscript for a final text editing.

Author Response

We have considered all issues mentioned in the reviewers’ comments and the whole manuscript was carefully revised and modified according to the reviewer´s suggestions. 

  • This study deals with a topic of interest. It discusses the intrinsic anthelmintic activity of carvone and demonstrates that its coadministration significantly increases the plasma bioavailability of IVM. Moreover, it exhibits carvone concentration profiles achieved in tissues and in target parasites for the first time. The manuscript is well written and i would like to suggest only some minor corrections:

Thank you very much for your comments.

  • in the abstract define the HPLC by which drug concentrations were measured in plasma, target tissues and in  contortus

It was defined.

  • spp is spp. and should not be in italics

It was corrected.

4) line 80 – delete full stop

-line 390 – correct

- line 407 – delete full stop

They were corrected.

5) Finally, please go through the manuscript for a final text editing.

The whole manuscript was revised.